# The *Arabidopsis* embryo as a quantifiable model for studying pattern formation

Yosapol Harnvanichvech[1,2,*] , Vera Gorelova[2,*], Joris Sprakel[1] and Dolf Weijers[2]

[1]Physical Chemistry and Soft Matter, Wageningen University, Wageningen, The Netherlands; [2]Laboratory of Biochemistry, Wageningen University, Wageningen, The Netherlands; * These two authors have contributed equally

cell specification; computational cell biology; gene expression; plant embryogenesis; plant development; pattern formation.

**Author for correspondence:**
Dolf Weijers,
E-mail: dolf.weijers@wur.nl

## Abstract

Phenotypic diversity offlowering plants stems fromcommon basic features of the plant body-pattern with well-defined body axes, organs and tissue organisation. Cell division and cell specification are the two processes that underlie the formation of a body pattern. As plant cells are encased into their cellulosic walls, directional cell division through precise positioning of division plane is crucial for shaping plant morphology. Since many plant cells are pluripotent, their fate establishment is influenced by their cellular environment through cell-to-cell signaling. Recent studies show that apart from biochemical regulation, these two processes are also influenced by cell and tissue morphology and operate under mechanical control. Finding a proper model system that allows dissecting the relationship between these aspects is the key to our understanding of pattern establishment. In this review, we present the *Arabidopsis* embryo as a simple, yet comprehensive model of pattern formation compatible with high-throughput quantitative assays.

## 1. Introduction

Flowering plants demonstrate a wealth of phenotypic diversity, but the structure of their body pattern with well-defined body axes, organs and tissue organisation is highly similar across different species (Jürgens et al., 1994). How such a stereotypical body pattern is achieved remains one of the most fascinating questions of developmental biology. Regulation of cell shape and identity is crucial in this process. Since plant cells are encased in cellulosic walls and are unable to move, proper morphology relies substantially on oriented cell division through adjustment of division plane positioning, so that daughter cells are placed in their specific locations (Smith, 2001). Apart from controlling cell topology, positioning of division plane also determines whether a cell divides symmetrically and produces daughter cells of similar sizes, or asymmetrically with daughter cells of different sizes (Scheres & Benfey, 1999). Asymmetric cell divisions, often considered formative divisions, are key to tissue formation, and require that resulting daughter cells not only differ in morphology, but most importantly possess distinct identities (Scheres & Benfey, 1999). Acquisition of different identities is preceded by establishment of cellular polarity reflected in segregation of subcellular components such as hormones, mRNA, proteins and organelles between two daughter cells, and can be influenced by both intrinsic and extrinsic mechanisms (Dong et al., 2009; Kimata et al., 2016; Kimata et al., 2019; Kimata et al., 2020). Intrinsic regulation relies on activation of cell type-specific transcriptional networks, while extrinsic control is determined by cellular environment and involves cell-to-cell communication through different mobile molecular signals such as short miRNA, short peptides, hormones, as well as mechanical cues (Heisler et al., 2010; Schlereth et al., 2010).

Plant morphogenesis and organogenesis are incredibly plastic (Dingkuhn et al., 2005; Ramage & Williams, 2002). Unlike in animals, those processes continue in plants in post-embryonic development and are actively modulated by environmental cues. In fact, the vast majority of plant morphological features is formed post-embryonically. However, the basic tissue pattern with main tissue types such as epidermis, vascular and ground tissue, is formed during embryogenesis and is only reiterated in post-embryonic development (ten Hove et al., 2015). Early plant embryo

development is not accompanied by the cell differentiation events that are widespread in post-embryonic development, such as root hair formation at the basal side of root epidermal cells, or suberin and lignin deposition in the root endodermis. This makes early embryogenesis a simple model allowing for dissection of pattern formation without the complication of cell differentiation pathways. Embryogenesis of most flowering plants is characterised by rather variable cell divisions – a trait highly undesirable for a model system intended for pattern formation studies (Johri et al., 2013; Johri & Ambegaokar, 1984; Pollock & Jensen, 1964). Only few species, including the best characterised plant model object *Arabidopsis thaliana*, show a highly regular division pattern (Jürgens et al., 1994; Mansfield & Briarty, 1991). Already at early stages of *Arabidopsis* embryogenesis comprising only 32 cells, the body axis and all major tissue types are established (ten Hove et al., 2015). The sufficiency of such a modest cell number together with the high predictability of its division pattern makes *Arabidopsis* embryogenesis an excellent model for studying plant body pattern formation. The versatility of this model was further cemented by development of embryo-optimised reporters of various subcellular structures, compartments and hormone response and establishment of cell fate-specific markers (Liao et al., 2015; Liao & Weijers, 2018).

While conventional approaches of mutant screening for defects in cell division and cell identity establishment revealed many important regulators of pattern formation (Benfey et al., 1993; Laux et al., 1996; Scheres et al., 1995), its efficiency and sensitivity is inherently very low, particularly in species that have large gene families. Moreover, perturbations of cell division during pattern formation might lead to embryo lethality that renders mechanistic studies rather challenging. Relatively recent application of high-throughput transcriptomic and proteomic approaches has significantly advanced our understanding of pattern formation in different developmental contexts (Brady et al., 2007; Levesque et al., 2006; Sozzani et al., 2010). The output of these studies was used to create publicly available databases and user-friendly tools that further facilitated elucidation of developmental processes (Le et al., 2010; Palovaara et al., 2017; Waese et al., 2017). Apart from enabling deeper elucidation of developmental pathways, this advancement underscored the importance of quantitative approaches and incentivised generation of sensitive tools allowing for accurate quantitative analysis of dynamics of subcellular components as well as cell morphology and mechanics (de et al., 2015; Liao et al., 2015; Liao & Weijers, 2018).

This manuscript aims to provide a current state of the art of the high-throughput and quantitative approaches that are presently used to shed light onto processes of plant body pattern formation using the early *Arabidopsis* embryo as a model. Furthermore, it outlines the current trends in the field and proposes directions for future research.

## 2. Quantification of cell morphology during pattern formation

The question of how a cell positions its division plane has been puzzling researchers for centuries. In the 19th century, scientists studying cell division in multicellular algae suggested that cells behave as soap bubbles and tend to minimise the interface between two daughter cells, such that cells divide along the shortest path (Errera, 1888). However, for a given cell geometry, several short routes exist. Therefore, a cell divides along one of the shortest paths (Besson & Dumais, 2011). This hypothesis was successfully applied

to explain cell divisions in the *Arabidopsis* shoot apex, however, it could not account for divisions in other contexts, particularly, where cells divide asymmetrically, for instance, during stomata development (Dong et al., 2009). The question of how cells 'break' the symmetry became one of the central questions of developmental biology.

To address this question, one should be able to recognise whether a cell divides symmetrically or asymmetrically. This task is not as trivial as it might appear, in large part, due to complex polyhedral shapes that plant cells can adopt. The early two-dimensional approaches employing classical histology could not always resolve this complexity and the question whether pattern formation during embryogenesis is accompanied by morphologically asymmetric divisions remained open (Bougourd et al., 2000). The recent advancement in three-dimensional (3D) imaging allowed to address this question and accurately characterise morphology of patterning during *Arabidopsis* embryogenesis (Figure 1) (Moukhtar et al., 2019; Truernit et al., 2008; Yoshida et al., 2014). This advance became an important milestone in pattern formation research and consolidated the *Arabidopsis* embryo as an excellent model for its elucidation. The comprehensive 3D analysis of *Arabidopsis* embryogenesis demonstrated that formative divisions that are accompanied by expression of fate-specific markers are morphologically asymmetric, that is generating daughter cells with different volumes (Yoshida et al., 2014). Examples where fate acquisition correlates with asymmetry in cell division are well known in post-embryonic development (De et al., 2008; Robinson et al., 2011). Interestingly, the analysis almost invariably identified embryos of discrete 2-, 4-, 8- and 16-cell stages, suggesting that cell divisions might be synchronised during embryogenesis.

Most importantly, the study provided the first insight into how cells might disobey the shortest path rule and 'break' the symmetry of division. Specifically, it demonstrated that mutant embryos, constitutively expressing an inhibitor of auxin response, indole-3-acetic acid inducible 12/bodenlos (iaa12/bdl), switch from morphologically asymmetric to symmetric divisions, suggesting that breaking the symmetry is an auxin-dependent process (Yoshida et al., 2014). A later study suggested that also the first asymmetric divisions can conform to a shortest path rule passing through the nuclear centroid, provided that newly inserted walls are curved (Moukhtar et al., 2019). Further, mechanistic studies will be required to determine the regulation of these divisions, and to address whether the same principle applies to post-embryonic development.

The accurate characterisation of cell morphologies during pattern formation was enabled by a powerful 3D image analysis tool, MorphoGraphX, that transfers recorded cell geometries into computational modelling environments (Figure 1) (de et al., 2015). This novel computational approach instantly found application in addressing diverse questions in the field of developmental biology (Ma et al., 2019; Sapala et al., 2018; Stanislas et al., 2018). Recently, a counterpart of this powerful tool, PlantSeg, employing principles of neural networks and allowing for high-throughput analysis of 3D images of more complex multicellular specimens, became available (Wolny et al., 2020).

These tools provide an opportunity to explore interactions between genes and morphology during pattern formation. Their compatibility with analysis of fluorescent reporters opened possibilities for dissecting the dynamics of subcellular organisation during polarity establishment and cell division. Assessment of cell geometries with MorphoGraphX combined with modelling approaches has been successfully applied to demonstrate a link

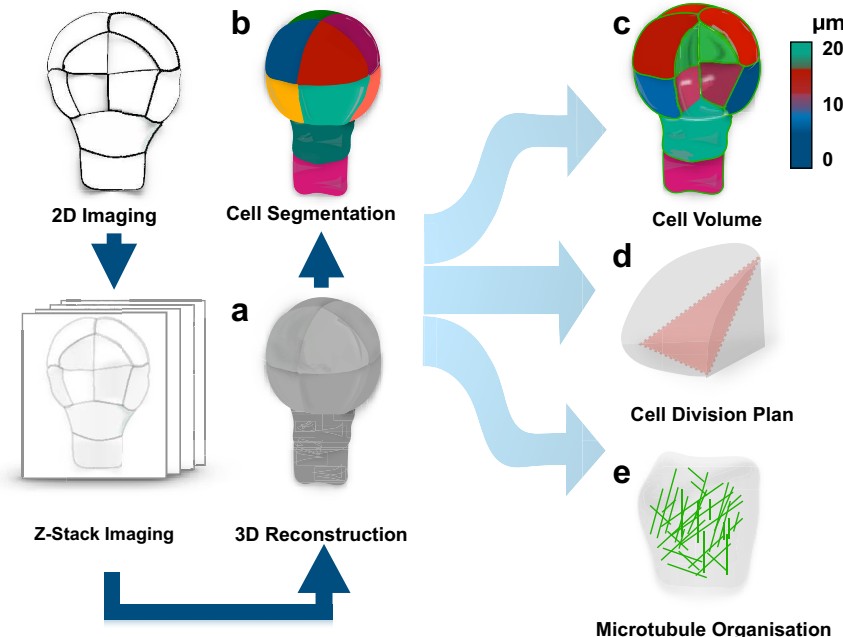

**Fig. 1.** Quantification of cell morphology during pattern formation. Data extracted from 3D reconstructed *Arabidopsis* embryogenesis (a) transferred into computational modelling environment using MorphoGraphX or PlanSeg (b) to accurately assess morphologies of individual cells (c), determine positioning of division planes (d) and trace fluorescently tagged subcellular structures (e). The data can be used in simulation modelling, allowing to trace interactions between shapes and cellular responses.

between arrangement of cortical microtubules and division plane positioning (Figure 1) (Chakrabortty et al., 2018).

However, thoroughly and comprehensively, the work by Yoshida et al. (2014) and Moukhtar et al. (2019) characterised embryonic cell divisions, it used fixed samples, and therefore lacked the bona fide time dimension. Hence, the real-time patterning process remains to be explored. A genuine 4D technique could be a powerful addition to the current approach and would allow, at the very least, clear up the presumptive synchronisation of cell division during embryogenesis. Addition of the real-time resolution would allow to reconstitute the sequence of events leading to asymmetric divisions and accurately track processes contributing to pattern formation. Achievement of these goals became feasible with application of an *in vitro* ovule cultivation system allowing tracking of early embryogenesis events (Gooh et al., 2015). Live-cell imaging was also successfully used to study mitochondria, vacuole and cytoskeleton organisation of *in vitro* cultivated *Arabidopsis* zygotes (Kimata et al., 2016; Kimata et al., 2019; Kimata et al., 2020). Although these live-imaging techniques provide a unique opportunity to visualise processes occurring in the embryo enclosed into developing seed, they are not without their own challenges. These approaches utilise highly expressed markers that might potentially interfere with normal developmental processes. Further improvement of the live-imaging approaches will increase value of the *Arabidopsis* embryo as a model system.

## 3. Quantification of biochemical aspects of pattern formation

To understand the regulation of cell division and to address the workings of the machinery that adjusts division plane positioning during pattern formation, one needs the capacity to determine which subcellular (molecular and morphological) events precede cell division. This will require a rather comprehensive approach that will allow to interrogate the cell for the earliest changes in its transcriptome, establishment of polarity and finally, the assembly of the

new cell wall. A key to tackling this problem is the development of highly sensitive methods that can track changes in gene expression at a single cell level and generation of robust quantitative reporters that could be monitored overtime. This part of the manuscript will discuss the state of the art, highlight current approaches, available reporters and tools for monitoring pattern formation.

## 4. The hunt for regulators and first events in cell specification

Identity of a cell can be inferred from its biochemical characteristics and morphology. Since modulation of transcription is one of the first cellular biochemical responses, one can view the emergence of specific expression profiles as the first step on the path to cell fate establishment. In order to elucidate how cell fate is established, researchers set out to find genes acting as regulators of pathways behind cell specification.

Initially, identification of genes acting as regulators of cell identity has been achieved using conventional methods, such as mutagenesis, promoter/enhancer trapping and genetic screening. Expression domains of the potential regulators have been determined using transcript localisation techniques including *in situ* hybridisation and the use of fluorescent reporters. This approach led to characterisation of many cell fate regulators (reviewed in ten Hove et al., 2015). Among them was *A. thaliana* meristem L1 layer (ATML1), a master regulator of shoot epidermal identity (Lu et al., 1996), MONOPTEROS (MP/ARF5), a determinant of hypophysis specification (Schlereth et al., 2010), HD-ZIP III genes involved in specification of the shoot apical meristem (SAM) (Prigge et al., 2005), PLETHORA (PLT) genes specifying root apical meristem (RAM) (Aida et al., 2004; Galinha et al., 2007), WUSCHEL (WUS), regulator of SAM specification (Laux et al., 1996). WUS-related homeobox (WOX) gene family was demonstrated to mark cell fate decisions during embryogenesis (Haecker et al., 2004). GRAS family transcription factor SHORTROOT (SHR) and its target SCARECROW (SCR), which is also a GRAS family member, were

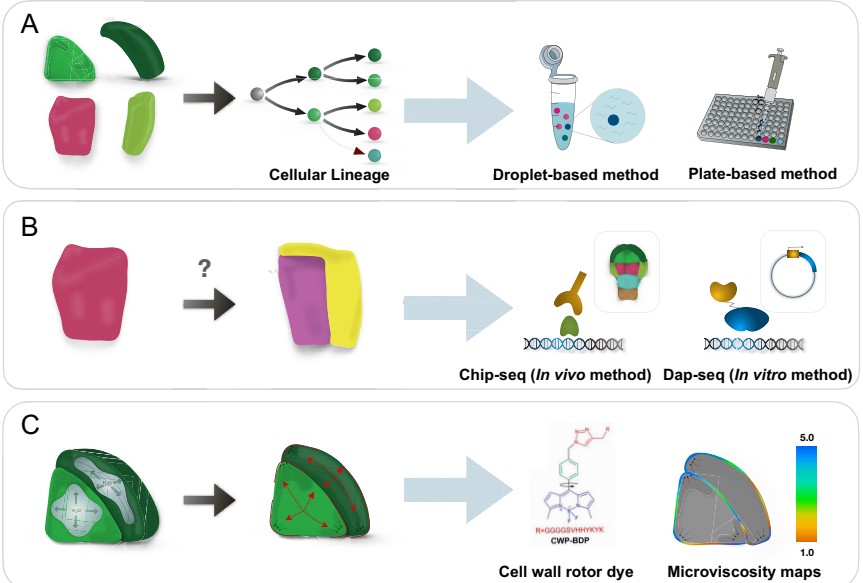

**Fig. 2.** Gaining insight into biochemical and mechanical responses during pattern formation in plants. (a) Study of cell identity establishment through generation of cell type-specific gene expression profiles. Following disintegration of embryos to single cells using lysis solution, cells are sorted according to the presence of fate-specific markers. The collected cells are further used in either droplet-based or plate-based single-cell RNA Seq assays that allow to interrogate cells for changes in transcriptome (b) Exploring cell-to-cell signaling during cell specification and pattern formation. A cell (pink) divides producing daughters with different fates (lilac and yellow). The molecular basis of fate specification is inferred using high-throughput approaches, such as Chip-seq and Dap-seq. (c) Visualisation of mechanical patterns in plants using mechano-probes. Cellular turgor pressure (gray shapes) causes tensile stress (red arrows) in the cell wall. The resulting tensile stress can be visualised and quantified using rotor dyes that change their fluorescence lifetime depending on the mechanical properties of the cell wall.

found to control ground tissue patterning. Specifically, SHR was shown to be required for establishment of endodermal identity and both SHR and SCR control asymmetric periclinal division of cortex/endodermis initial (Benfey et al., 1993; Levesque et al., 2006; Scheres et al., 1995). Although these conventional approaches helped to reveal a number of important regulators of cell identity, their depth and efficiency was rather limited. Genetic screening is not an ideal approach when it comes to the hunt for regulators of cell identity establishment, since mutant for their genes might be embryo lethal. At the same time, the search for master regulators might also be hampered by genetic redundancy inherent to plant genomes.

A deeper understanding of embryonic cell specification could be attained by generation of cell type-specific whole-genome expression profiles using high-throughput techniques, such as microarray and RNA-seq technologies (Figure 2a). Such high-throughput approaches have been used in the past to obtain cell type-specific transcript profiles of post-embryonic roots (Birnbaum et al., 2003; Brady et al., 2007). Owing to the facile breakdown of the root tissues to individual cells, these attempts proved very successful. Application of a similar tissue dissociation technique to embryos that are encased in developing seeds, however, turned out to be a challenging enterprise. First attempts involved expression profiling of whole seeds at different developmental stages (Girke et al., 2000; Le et al., 2010).

Generation of expression profiles of whole isolated embryos, liberated from the endosperm and the seed coat, at different developmental stages involved labour intensive excision of plant tissue using laser capture microdissection (Belmonte et al., 2013). This technique has been recently used to compare transcript profiles of apical and basal cell lineages of the early proembryo (Zhou et al., 2020). Fluorescence-activated nuclear sorting was used to separate fluorescently labelled nuclei from crude embryo preparations to generate transcript profiles of the early proembryo and the suspensor (Slane et al., 2014). However, this approach allows to analyse only nuclear transcripts and disregards those stored in the cytosol. To capture the cytosolic transcripts, that is actively translated or stalled on ribosomes, translating ribosome affinity purification has been used to generate expression profiles of plant tissues (Mustroph et al., 2009; Zanetti et al., 2005). This technique employing bead-based affinity purification of FLAG-tagged ribosomal protein L18 driven by cell-type specific promoters allowed to assess expression profiles of *Arabidopsis* seedlings with high spatial resolution (Mustroph et al., 2009).

Isolation of nuclei tagged in specific cell types (INTACT) circumvents this issue by utilising a two-component transgenic labelling system where biotin ligase (BirA) biotinylates a nuclear envelope-localised green flourescent protein [nuclear targeting fusion protein (NTF)] when co-expressed in the same cells. Biotin-tagged nuclei from crude plant tissue preparations are further isolated using streptavidin-coated beads. This technique was applied to map expression dynamics with cell type resolution at different stages of embryogenesis (Palovaara et al., 2017). But again, INTACT allows to explore nuclear transcripts and remains blind to cytosolic ones. Significant difference between nuclear and total cellular transcript profiles suggested that nucleus-specific approaches might potentially overlook important elements of cell fate specification (Palovaara & Weijers, 2019). Therefore, future efforts should seek to achieve generation of cell type-specific total cellular transcript profiles. Another important weakness of the abovementioned approaches is their reliance on cell type-specific markers that are often missing for certain cell types during various stages of embryogenesis.

Moreover, the availability of the gene expression data boosted establishment of important molecular markers allowing for tracking specification processes in the root stem cell niche and

vascular tissues (Smit et al., 2020). Collectively, embryonic tissue- and cell-specific gene expression datasets became an invaluable resource for studying establishment of cell identity. Development of tools providing graphical representation of the embryonic gene expression profiles such as the *Arabidopsis* ePlant browser (Waese et al., 2017) (http://bar.utoronto.ca/eplant/), SeedGene Network (Le et al., 2010) (http://seedgenenetwork.net/) and AlBERTO (Palovaara et al., 2017) (http://www.albertodb.org) made these powerful data sources easily assessable for the whole community of plant researchers.

Development of technologies allowing to infer cellular identity is undoubtedly an important milestone in developmental biology. In the future, these techniques will enable us to precisely track the establishment of cellular fates and will prove valuable in addressing the mechanisms underlying acquisition of cell identities.

## 5. Eavesdropping on cell-to-cell communication

Since most plant cells are pluripotent, fate establishment is profoundly influenced by the cellular context (Scheres, 2001; Serna et al., 2002). Therefore, cell-to-cell communication is an essential aspect of cell specification and pattern formation in plants. To date, several key molecular cues involved in transmission of signals between cells have been identified. The most prominent among these molecules is the phytohormone auxin that modulates gene expression by promoting degradation of auxin/indole-3-acetic acid (Aux/IAA), transcriptional repressor proteins, leading to release of auxin response factors (ARFs) transcription factors, which are otherwise bound by Aux/IAA repressors. Specifically, auxin mediates binding of Aux/IAA to TIR1/AFB, a part of SCF (SKP1-CUL1-F box) complex, which results in ubiquitination and subsequent degradation of Aux/IAA. Released ARFs bind to auxin response elements (AuxREs) to activate or inhibit target gene transcription (Overvoorde et al., 2005; Tiwari et al., 2004; Ulmasov, Hagen, et al., 1997; Ulmasov, Murfett, et al., 1997; Villalobos et al., 2012). Auxin is a mobile signal that is actively transported from cell to cell by plasma membrane localised AUXIN1/LIKE-AUX1 (AUX/LAX) and PIN-FORMED (PIN) proteins that facilitate its influx and efflux, respectively (Bennett et al., 1998; Friml, 2003; Friml et al., 2003; Gälweiler et al., 1998; Marchant et al., 1999; Petrášek et al., 2006; Swarup et al., 2001). The efflux transporters are polarly localised, assuring the directionality of the auxin flux through the plant body (Petrášek et al., 2006; Petrášek & Friml, 2009). This polar transport integrates cellular and tissue polarity and underlies a multitude of developmental processes in plants (Křeček et al., 2009; Petrášek & Friml, 2009). Interference with either biosynthesis, transport or response strongly interferes with normal development (Cheng et al., 2007; Robert et al., 2015; Stepanova et al., 2008).

Auxin is profoundly involved in pattern formation and cell specification in diverse developmental contexts in *Arabidopsis*: root pattern formation (Sabatini et al., 1999), asymmetric division of the cortex/endodermis initial in post-embryonic development (Cruz-Ramírez et al., 2012), as well as formative divisions during lateral root formation (Bishopp, Help, et al., 2011; Bishopp, Lehesranta, et al., 2011; De et al., 2007; Moreno-Risueno et al., 2010). Similarly, pattern formation during embryogenesis critically depends on auxin signaling (Hardtke & Berleth, 1998; Jürgens, 2001; Schlereth et al., 2010; Yoshida et al., 2014). Thus, auxin is involved in formative division in vasculature formation during embryogenesis (De et al., 2013; Ohashi-Ito et al., 2013).

The *Arabidopsis* genome encodes 6 TIR/AFB, 29 Aux/IAA and 23 ARF proteins (Weijers & Wagner, 2016). Differential TIR/AFB–Aux/IAA and Aux/IAA–ARF interactions likely contribute to the complexity and diversity of auxin responses (Piya et al., 2014; Villalobos et al., 2012). Moreover, differential expression of ARFs during embryogenesis, observed in the study using fluorescent reporter lines, suggested that auxin response machinery might be adjusted to elicit cell type-specific transcriptional responses (Rademacher et al., 2012). Information of transcription factor and DNA binding interaction from Chromatin immunoprecipitation followed by sequencing (ChIP-seq) (Park, 2009) or DNA affinity purification sequencing (DAP-seq) (Bartlett et al., 2017) in combination with high-resolution single cell expression profiles will help to gain a deeper insight into workings of the auxin response machinery (Figure 2b).

Auxin dependent cell-to-cell communication is mediated not only through the transport of auxin, but also through intercellular movement of its downstream effectors. Auxin response factor5/monopteros (ARF5/MP) expressed in the embryo proper drives hypophysis specification, thereby regulating non-cell autonomous cell fate establishment. Target of MP7 (TMO7), which expression is activated by ARF5/MP in the embryo, mediates this regulation by moving from the embryo proper to the hypophysis precursor (Schlereth et al., 2010).

The ability to track auxin signaling through visualisation of transcriptional response was crucial for the studies exploring the role of auxin in cell-to-cell communication and pattern formation. This was first enabled by the use of a synthetic auxin response reporter DR5 consisting of 7–9 AuxRE repeats (ARF binding sites) and marking sites of transcriptional auxin response by activating fused reporter genes such as $\beta$-glucuronidase, fluorescent proteins or luciferase (Friml et al., 2003; Moreno-Risueno et al., 2010; Ulmasov et al., 1997). However, the DR5 reporter failed to visualise computationally predicted auxin gradients and therefore was referred to as 'auxin maxima' reporter (Grieneisen et al., 2007). More sensitive visualisation of auxin responses was achieved by introduction of a new reporter, DR5v2, harbouring higher affinity ARF binding sites ( Liao et al., 2015). However, the most prominent advancement was achieved by development a two-component ratio-metric auxin response reporter, R2D2, that allowed for the most accurate visualisation and a semi-quantitative analysis of auxin readout in both embryonic and post-embryonic development (Liao et al., 2015). This quantitative reporter has been successfully used to visualise auxin response in specific embryonic tissues, such as vasculature, ground tissue, protoderm, hypophysis and suspensor (Möller et al., 2017; Roodbarkelari et al., 2015; Smit et al., 2020).

Besides auxin, several other phytohormones have been shown to be involved in pattern formation during embryogenesis. Thus, antagonistic interaction of auxin with cytokinin was demonstrated to control embryonic root stem-cell niche establishment. The study employed two-component-output sensor fused to luciferase to visualise cytokinin signaling (Müller & Sheen, 2008). Such mutually inhibitory activity of these two hormones was detected in pattern formation and cell specification in other developmental contexts outside embryogenesis (; Bishopp, Help, et al., 2011; Chang et al., 2013; Mähönen et al., 2006). It is conceivable that the interaction of hormonal cues might be used to couple positional information with fate establishment. In order to address this possibility, one has to be able to track and quantify the readout of the hormonal cues, which presently remains an obstacle due to the lack of proper tools (other than auxin-specific ones). A novel set of synthetic and modular hormone activated Cas9-

based repressors (HACRs) reporters allowing for visualisation of auxin, gibberellin and jasmonates, and hold great promise to overcome this obstacle (Khakhar et al., 2018). HACRs include deactivated Cas9 (dCas9) protein fused to a highly sensitive hormone-responsive degron domain and a part of the TOPLESS transcriptional repressor. Association of dCas9 with a guide RNA targets HACR to a promoter (such as *pUBQ1*), leading to transcriptional repression of a reporter (nuclear targeted Venus-Luciferase). Upon hormone accumulation, the degron sequence targets the HACR for ubiquitination and subsequent proteasomal degradation. Thus, in parallel to the natural hormone response, hormone accumulation triggers relief of repression on HACR target genes (reporter).

The system was transformed to *A. thaliana* that were genetically engineered for consistently producing a fluorescent protein. When there was no hormone, HACRs repress fluorescent gene which is consequent to an absent of fluorescent signal. If the hormone is present, HACRs is degraded and therefore fluorescent signal is present. With the ability of reducing fluorescent signal at different time point, HACRs system can finely detect hormonal level and track re-programming in developing plants.

Since cells need to transmit different types of information to their neighbours, for example, topology, stress, identity, differentiation state, one might expect the means of cell–cell communication to be rather sophisticated and involve different types of signals to avoid any miscommunication. In this light, small non-coding RNA molecules such as microRNA (miRNA) have emerged as an important signaling element, mediating cell–cell communication in plant development. Embryonic miRNAs were demonstrated to mediate regulation of gene expression by modifying the stability of embryonic messenger RNA, and thereby coordinating cellular transcript profiles during development. Thus, miR165/166 were shown to be important for establishment of shoot apical meristem at the globular stage of embryo development by repressing HD-ZIP lll genes (Müller & Sheen, 2008; Vashisht & Nodine, 2014). Furthermore, miR165/166 were also shown to be expressed in the basal-peripheral region, and were suggested to non-cell autonomously control expression of HD-ZIP III genes in specification of RAM during embryogenesis (Miyashima et al., 2013). Another type of miRNA, namely miR394, is required to maintain apical meristem and stem cell niche by repressing putative F-box protein (leaf curling responsiveness) (Song et al., 2012). Using fluorescent reporter lines, miR394 was found to first appear in the most outer layer of shoot apex (L1) and at later stages extend its expression to the inner layer (L2 and L3), suggesting a role of this miRNA in maintenance of three niches, including protoderm (in L1), ground meristem (in L2) and procambium (in L3).

Comprehensive analyses of miRNA distribution and dynamics in embryogenesis provided an opportunity to gain a deeper insight into the roles of these molecules in pattern formation (Plotnikova et al., 2019; Seefried et al., 2014). The study by Plotnikova and co-workers suggested that over 300 miRNAs operate during embryogenesis, and identified 59 high-confidence miRNA targets (Plotnikova et al., 2019). Previously, similar large-scale miRNA datasets have been used in development of user-friendly tools, such as miRBase (Kozomara & Griffiths-Jones, 2014) (http://www.mirbase.org/), psRNATarget (Dai et al., 2018) (http://plantgrn.noble.org/psRNATarget/) and plant microRNA database (Yi et al., 2015) (http://structuralbiology.cau.edu.cn/PNRD). These resources hold a great potential to advance our perception of the roles of miRNA in pattern formation during embryogenesis.

However, profound understanding of how plant cells are specified during pattern formation requires higher resolution that can be gained through generation of single cell expression profiles using advanced techniques, such as assays employing a droplet-based platform Drop-seq (Macosko et al., 2015) or plate-based platform CEL-seq (Hashimshony et al., 2016), Seq-well (Gierahn et al., 2017) and SMART-seq (Picelli et al., 2014) (Figure 2a). Specifically, SMART-seq and CEL-seq offer methodologies an efficient way to profile low number of single cells by sorting cells into microwells or plates, that host all the steps from the cell lysis to construction of cDNA libraries (Hagemann-Jensen et al., 2020; Hashimshony et al., 2016; Picelli et al., 2014). Other approaches such as Drop-seq and inDrop utilise separation of cells through encapsulating individual cells into nanoliter-sized droplets, followed by collecting all samples in a single tube. With few handling steps, the droplet-based approaches are suited for profiling of large numbers of cells at once (Macosko et al., 2015; Zilionis et al., 2017). In comparison with droplet-based techniques, plate-based assays allow for a greater flexibility in terms of discrimination of individual cell samples (individual wells) by quality. Moreover, they enable interruption of the assay or long-term storage of samples at ultra-low temperatures. With these advanced scRNA-seq technologies, generation of highly accurate single-cell transcript profiles of the first steps of plant development is within reach. However, the outcome and the current enigma of the cell type-specific adjustment of molecular composition is the positioning of division plane, that will ultimately contribute to pattern formation.

## 6. Digging into the cell interior

Positioning of division plane during asymmetric divisions is crucial for pattern formation in plants. However, the cues and the machinery that adjust division planes remain to be understood. Asymmetric divisions are preceded by the establishment of cellular polarity that is reflected by the segregation and organisation of subcellular components, such as hormones, mRNA, proteins and organelles (Dong et al., 2009; Kimata et al., 2016; Kimata et al., 2019).

Specific dynamics of cytoskeletal elements, namely actin and microtubules, was shown to precede the first asymmetric zygotic division in *Arabidopsis* (Kimata et al., 2016). Additionally, actin dynamics has been suggested to be important for polar auxin transport (Dhonukshe et al., 2008). Microtubule organisation was shown to strongly correlate with PIN1 localisation in the *Arabidopsis* shoot apex (Heisler et al., 2010). Microtubules were also reported to guide deposition of cellulose fibers, thereby affecting the direction of cellular growth (Bringmann et al., 2012; Paradez et al., 2006). Both microtubules and actin are required for early events in lateral root formation (Barro et al., 2019). Spatial organisation and dynamics of other organelles and compartments are also important to consider. For example, polar vacuole distribution is essential for accurate zygotic division, while nuclear dynamics and migration are important for asymmetric divisions during stomata formation (Kimata et al., 2019; Muroyama et al., 2020) and lateral root formation (De et al., 2010; Vermeer et al., 2014). Activity of endomembrane trafficking is essential for polar localisation of auxin transporters (Doyle et al., 2015; Kleine-Vehn et al., 2006). So far, the roles of dynamics and organisation of subcellular structures in asymmetric divisions and auxin transport, were studied using either *Arabidopsis* zygote of post-embryonic organs, and until recently were completely overlooked in embryogenesis. Tackling

these questions using the model of the *Arabidopsis* embryo with its highly stereotypical division pattern would allow to accurately link subcellular dynamics and division plane positioning.

A recently developed collection of embryo-optimised fluorescent subcellular reporters now allows to tackle this question. These reporters, named *Arabidopsis* cellular markers for embryogenesis (ACE), are based on a set of available markers for various subcellular structures driven by embryo-specific promoters such as *RPS5A* and *WOX2*, allowing to track their organisation and dynamics (Liao & Weijers, 2018). Actin and microtubule reporters from this series, namely ACE-14 and ACE-15, allowed to reveal the distinct organisation of the cytoskeletal elements at different stages of embryogenesis. Reporters for the Golgi complex and endosomal elements, GOT1p (ACE-09) and RabC1 (ACE-07), respectively, enabled tracing early secretion events. Plasmodesmatal marker PDCB1 (ACE-13), allowed visualisation of patterns of plasmodesmata channels essential for passive transport and cell-cell communication. ACE-11 reporter marking nuclear pore complex could help to address whether nuclear dynamics is also important for formative divisions during embryogenesis.

The importance of organisation of subcellular structures in asymmetric divisions was strongly emphasised by a study employing live-time imaging to trace the cytoskeleton in zygote polarisation and division (Kimata et al., 2016). This work highlighted the necessity to include the time dimension in studies elucidating establishment of polarity and pattern formation. The current knowledge of patterning during embryogenesis was gained from analysis of fixed samples and the time dimension was included by screening of a large amount of fixed samples, so that the whole duration of early embryogenesis was monitored (Yoshida et al., 2014). Addressing how embryonic patterning occurs in real-time became a reality with development of *in vitro* ovule culture system that allows normal ovule growth and embryo development from zygote combined with live imaging (Gooh et al., 2015).

## 7. Probing cell mechanics

For decades regulation of cell division plane positioning in plants has been attributed to biochemical cues (Pillitteri et al., 2016). Compelling evidence suggests that mechanical stimuli are as important as biochemical cues in instructing division plane orientation and regulating key processes in plant development, such as differentiation, expansion, cell fate specification and polarity (Heisler et al., 2010; Louveaux et al., 2016; Nakayama et al., 2012; Sampathkumar, Krupinski, et al., 2014; Sampathkumar, Yan, et al., 2014). However, what is the origin of mechanical forces in plant tissues?

Plant cell walls experience tensile stress caused by cellular turgor pressure. To resist this stress, cell walls alter their mechanical properties by depositing cellulose microfibrils and other components, such as pectins and xyloglucans (Braybrook & Jönsson, 2016). Since a cell shares its cellulosic walls with its neighbours that might have different growth rates and direction, mechanical properties of a single cell might be highly anisotropic. Such mechanical anisotropy reflects stress patterns present in tissues and instructs positioning of division plane. This phenomenon was first demonstrated on the model of the *Arabidopsis* shoot apex (Nakayama et al., 2012). It has been noted that in its central region, where stress is isotropic, cells divide along the shortest path (symmetrically), following Errera's rule. However, in its boundary region, where stresses are highly anisotropic, Errera's rule is broken and cells divide along the max-

imal tensile stress, which is a longer path (Louveaux et al., 2016). Corroborating the notion that cell divisions are under mechanical control, cell ablation led to re-arrangement of stress pattern and resulted in re-orientation of cell divisions in the shoot apex (Heisler et al., 2010). The mechanical stimulus in this experiment has been translated into subcellular response, resulting in re-polarisation of the auxin transporter PIN1, and re-organisation of microtubules. Such profound effect on cellular polarity in response to mechanical stimuli is expected to be accompanied and/or preceded by changes in gene expression profiles. A number of genes which expression is under mechanical control has been identified. Most prominent among those are TOUCH genes (Braam, 2005; Hamant & Haswell, 2017; Landrein et al., 2015). To understand how mechanical control of cell division is executed, one has to address following questions: (a) how mechanical stimuli are perceived and (b) translated into biochemical responses; (c) what is the chain of subcellular events that leads to adjustment of division plane in response to mechanical cues?

Cells can perceive mechanical signals through activation of mechanosensitive ion channels. Upon activation, such channels allow for ion passage down the electrochemical gradient (Hamant & Haswell, 2017; Hamilton et al., 2015; Peyronnet et al., 2014). Interestingly, a recent study demonstrated that blocking calcium channels prevents PIN1 re-polarisation and organogenesis in the SAM, corroborating the notion that ion fluxes are located upstream of establishment of polar protein domains in mechanotransduction (Li et al., 2019).

Recently, cortical microtubules started to emerge as possible sensory elements of the cellular interior. The capacity of microtubule arrangement to respond to mechanical stimuli and to correlate with tissue stress patterns allowed to consider this structures as a proxy for mechanical stress distribution (Hamant et al., 2019; Heisler et al., 2010; Louveaux et al., 2016; Sampathkumar, Krupinski, et al., 2014). A recent study demonstrated that organisation of microtubules determined by cellular geometry instructs cell division (Chakrabortty et al., 2018). This suggests that the feedback between geometry and mechanics might be mediated through these cytoskeletal elements (Chakrabortty et al., 2018; Hamant et al., 2019). To gain a deeper insight into regulation of this interaction and to shed light onto translation of mechanical stimuli into cellular responses we need to be able to trace subcellular components with high spatial and temporal resolution.

To understand how mechanical cues shape plant development, one has to be able to determine tissue stress patterns. Our current understanding of mechanical stress patterns in plant tissues has been gained from studies using atomic force microscopy, measuring stress experienced by the cell wall. This approach allowed to determine stress patterns of the epidermal cell layer of SAM, leaf pavement cells and hypocotyl epidermal layer (Peaucelle et al., 2015; Sampathkumar, Krupinski, et al., 2014). However, this technique could not be applied to probe mechanical properties of deeper tissue layers, leaving questions about the role of mechanical stimuli in pattern formation unanswered. Another question that remains to be addressed is how mechanical stimuli from the cell wall are translated into subcellular response. As the cell wall is separated from the cellular interior by the plasma membrane, it is tempting to speculate that mechanical cues are translated into properties of that barrier compartment. The recent development of sensitive cell-permeable mechano-probes that allow to visualise tension patterns of the plasma membrane and porosity of the cell wall provided an excellent opportunity to fill this knowledge gap and to visualise the stress patterns of inner tissues of

plant organs (Figure 2c) (Michels et al., 2020). Combined with reporters for subcellular structures and tools for 3D image analysis such mechano-probes provide an excellent opportunity to dissect the relationship between mechanical stress patterns, morphology and biochemical responses and to reveal the feedback regulation between these elements at cellular, tissue and organ levels.

## 8. Application of modelling approaches in elucidation of pattern formation

Experimental molecular studies, discussed in the previous sections, typically focus on individual elements encoded by a genome or small arrays of elements to derive their interaction and to explore their contribution to biological processes. Since such studies can only deal with a limited number of elements at a time, they are restricted in power. In part, this stems from the fact that molecular pathways orchestrating biological processes are multicomponent, nonlinear and involve complex feedback regulatory loops. The large number of different input signals controlling a single biological process further adds up to the complexity. This renders experimental molecular approaches unable to provide comprehensive insights into the complex machineries driving biological processes. For instance, as shown in previous sections, a process of cell division is instructed by biochemical signals involving transcription factors, miRNA, hormones, and so on, being at the same time under mechanical control and dependent on cellular environment and cell and organ geometry.

Development of mathematical modelling approaches empowered researchers to tackle such complex problems and to test contributions of individual elements and input signals and their interactions to an output trait. Theoretical approaches also enable prediction of regularities and processes that cannot be tested or visualised due to certain technical limitations. Thus, although accurate experimental visualisation of auxin distribution was not possible, computational studies enabled its modelling and pinpointed the importance of polar auxin transport in the root tip and emphasised its role in the RAM establishment and root zonation (Grieneisen et al., 2007; Mironova et al., 2010), predicted role of AUX/LAX influx carriers and PIN efflux transporters in control of auxin distribution in the root tip (Band et al., 2014) that was further experimentally supported on the model of *Arabidopsis* embryo (Robert et al., 2015). Studies coupling auxin distribution predictions with expression profiles demonstrated that auxin influx and efflux coordinate gene expression during lateral root development (Péret et al., 2013).

The use of gene expression data for the modelling of genetic networks has long been exercised to explore various aspects of plant development and physiology (De, 2002). Specific focus on the modelling of genetic networks was laid in studies describing pattern formation and identity specification. Thus, gene expression visualised by confocal imaging was used to develop a computational model of bisymmetric patterning of the root vascular system predicting interplay of SHORTROOT, miRNA165/6, PHABULOSA, auxin transport and cytokinin signaling (Muraro et al., 2014). Importance of auxin-cytokinin interaction in vascular tissue patterning was further supported by De et al. (2014). The study identified a genetic network underlying an auxin-driven cytokinin biosynthesis in the early development of the *Arabidopsis* embryo and ensuring a correct vascular tissue patterning. Modelling study using live imaging data combined with gene expression profiling

and chromatin immunoprecipitation revealed importance of transcriptional repression of WUS and a set of transcription factors to prevent premature differentiation of SAM proliferation zone (Yadav et al., 2013). Previously confocal imaging data were used to model WUS expression domain in SAM (Jönsson et al., 2005). Modelling approaches also predicted various aspects of epidermal cell lineage, including root hair and trichome patterning and stomata development (Ryu et al., 2013). A recent study using the model of the *Arabidopsis* floral meristem and employing high-resolution time-lapse imaging provided an insight into the link between gene regulation and morphogenesis (Refahi et al., 2021).

Modelling studies covered various aspects of plant development at the organ and organismal levels, describing cell division patterns during lateral root formation (von et al., 2016) and establishment of apical–basal axis during embryogenesis (Wabnik et al., 2013). Theoretical studies also aimed at linking gene expression networks with organ shape. Thus, a computational model has tested the significance of the reciprocal repression between CUC2 and the PIN1 module in the determination of leaf margin shape (Bilsborough et al., 2011). Relatively recent advance of 3D imaging techniques enabled development of models explaining the relationship between cell shape, dynamics of subcellular structures and division plane orientation. Thus, orientation of the cortical microtubule arrays was suggested to determine positioning of division planes during embryogenesis (Chakrabortty et al., 2018). Another modelling study on embryo development suggested the possibility that cell divisions are guided by nuclear positioning (Moukhtar et al., 2019), drawing parallel with the role of nuclear movement in asymmetric divisions during post-embryonic development (De et al., 2010; Kimata et al., 2019; Muroyama et al., 2020; Vermeer et al., 2014). However, since the nucleus occupies a substantial part of the cellular volume in early embryo cells (Liao & Weijers, 2018), it remains to be investigated if and how regulated nuclear position can contribute to division orientation in these cells.

The dynamic interplay of geometrical and genetic inputs has attracted significant interest during the past decade. In this light, a novel modelling framework, morphodynamics, has been developed to explain complex temporal and spatial interactions of growth and signaling (Bassel et al., 2014; Jönsson et al., 2012). The recently emerged mechanical aspect of plant development was incorporated into computational studies to explain positioning of division plane orientation, cytoskeleton behaviour and polarity of stomatal stem cells (Louveaux et al., 2016; Sampathkumar, Yan, et al., 2014).

A fundamental progress over the past two decades is the link between computational modelling, molecular biology and advanced morphology analysis. Coupled with the recently developed approach of visualising tissue mechanical patterns (Michels et al., 2020), this framework holds a great potential in achieving a more comprehensive perception of pattern formation in plants (Figure 2).

## 9. Conclusions and future perspectives

During the past decade the *Arabidopsis* embryogenesis emerged as a versatile model allowing to address many aspects of plant development. Its low cell number and the stereotypical division pattern rendered the *Arabidopsis* embryo model to be simple enough for the use in modelling studies, while the presence of all basic tissue types and organs made this system sufficient to explore cell

specification and establishment of the body pattern. The recent development of embryo-optimised reporters and the advance of imaging techniques further secured the embryo in the role of a good model system.

However, as any model, *Arabidopsis* embryo system has its limitations. Specifically, its location inside the developing seed made the use live imaging approaches challenging until this issue was countered by the development of *in vitro* ovule culture systems (Gooh et al., 2015).

Despite these challenges, the use of *Arabidopsis* embryo model holds potential for establishing new methodologies. A typical example is a combination of the existing reporters for subcellular structures with the recently developed mechano-probes that emerged as an excellent tool capable of addressing how mechanical cues instruct cell division and pattern formation. Future development of high sensitivity techniques that would allow to trace subcellular components at high resolution in real time and techniques enabling single molecule imaging would help to accurately reconstruct the sequence of intracellular events leading symmetry breaking during cell division.

Finally, its highly predictable division pattern makes the *Arabidopsis* embryo a perfect model to address the question of symmetry breaking. The study by Yoshida et al. (2014) demonstrated the robustness of this system and uncovered auxin response as a requirement for symmetry breaking. It still remains to be determined if the same principle operates in other developmental (post-embryonic) contexts and a general question arises: How far we can extrapolate principles applying to the embryo model? We, therefore, anticipate the future research to focus on comparing different developmental contexts of the same species as well as cognate contexts of different species as it appears to be the final obstacle for *Arabidopsis* embryo to enjoy the most coveted feature of any model system – universality.

## Acknowledgements

The authors thank members of the Weijers and Sprakel labs for inspiring discussions.

**Financial support.** This work was financially supported by the Netherlands Organisation for Scientific Research (NWO; ECHO grant 711.018.002 to J.S) and the European Research Council (ERC; Advanced Grant 'DIRNDL'; Contract number 833867 to D.W).

**Conflict of interest.** The authors declare no conflicts of interest.

**Authorship contributions.** Y.H. and V.G. prepared the first draft of the manuscript and figures. J.S. and D.W. provided input to the draft and all authors contributed to writing the final version.

**Data availability statement.** No new data or code are presented in this paper.

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
