## [Reviewer Report]

Dear Olivier,

It is my pleasure to submit our invited review paper on the plant embryo as a model for quantitative plant biology. In our review, we have attempted to provide a forward-looking view on how this developmental system can be used to address questions related to the mechanisms underlying multicellular plant development. The review is over the 5000 word limit, and we welcome any feedback on whether you would like us to shorten the text.

We look forward to your feedback..

Kind regards,

Dolf

---

## [Reviewer Report]

*Comments to Author*: The manuscript by Harnvanichvech et al (“The Arabidopsis embryo as a quantifiable model for studying pattern formation”) reviews the current knowledge of the advantages and limitations on using Arabidopsis embryos as a working model to study the formation of patterns. It includes the different methodological approaches that can be used in patterning studies with up-to-date references. It covers molecular and biochemistry analysis of biological pathways (functional genetics, proteomics, transcriptomics [including cell-specific approaches]) imaging and modelling.

Overall the text is well structured and comprehensive for non-specialist readers. I would only comment here on the use of trivial abbreviations that could be explained at first use for naïve or non-plant readers (notably RAM, SAM, MT). In this direction, the legend of figure 2 may be more explanatory. It is now very succinct and a bit cryptic. The same in the text when referring to Fig 2A on page 12, the authors do not explain what is SMART-seq, Drop-seq, Cell-seq or Seq-well. Just dropping the names. What are they and why these techniques could be beneficial to studies in embryos?

Overall, the authors do not emphasize enough (in my opinion) that the embryos are embedded in seed coat and surrounded by endosperm (only mentioned in conclusion). Hence the technical challenges to use embryos as a model. This is notably important for cell-type transcriptomics where cell-type embryonic markers are necessary and should not be expressed in the seed coat, endosperm and eventually funiculus. This could be circumvented by embryo isolation. The same is true for imaging. The authors cite Gooh et al 2015, where seed culture was used to image embryos up to 16-cell stage. Yet, for this technique, markers need to be highly expressed (does that influence normal processes?). After this stage, notably for studies on vascular development or SAM development, seed coat with the presence of tannins and cuticle is problematic to live imaging as these tissues absorb emitting light in CLMS, for example.

Some minor comments, AUX/LAX proteins are not strictly polar proteins. In fact, these proteins are polar in few tissues. So, page 9, stating that PIN1 and AUX/LAX localize to opposite sides of the cells may be restrictive.

On page 11, the authors mentioned hormone tools. What about the HACRs? Do you think they may be useful tools?

Khakhar, A. et al., 2018. Synthetic hormone-responsive transcription factors can monitor and re-program plant development. eLife, 7, p.e34702.

---

## [Reviewer Report]

*Comments to Author*: This review, entitled "The Arabidopsis embryo as a quantifiable model for studying pattern formation", takes stock of the current state of knowledge on the subject as well as the main directions in which modern approaches of integrative biology are moving.

This document also gives an account of the remarkable progress made in this field (and notably by the research team that is the author of this review), particularly on the methodological tools used to quantify biological phenomena.

Overall, this review is well written and only minor changes need to be made before publication.

1) One can regret the lack of connection between the different parts which could be illustrated in the form of a major biological question; for example, what are the events (on different scales) that contribute to the establishment of a new identity?

2) In the introduction, the sentence "...proper morphology relies chiefly..." should be qualified: differential cell growth plays a significant role in morphogenesis.

3) In the introduction, "Acquisition of different identities..." this sentence would require a bibliographical reference to support the point.

4) In the introduction, " ...chaotic cell divisions..." I don't understand what the word chaotic means here, does the author mean variability? or chaos?

5) P.4 I doubt that the publication (Bougourd et al. ,2000) will allow a 3D quantification of cells. It is rather necessary to refer to (Truernit et al., 2008) protocol used in (Yoshida et al., 2014).

6) P.4 The cell division in the first generations is not synchronous, it is enough to increase the number of embryos to realize this.

7) P.4 It is not really a question of resolution in time but rather in space (number of cells) which allows us to infer time

8) P.4 Although the publication of (Yoshida et al., 2014) is indeed a pioneer in the 3D morphological analysis of the embryo, the fact remains that the first asymmetrical divisions follow a "shortest path rule" (Moukthar et al., 2019).

9) P17 the reference (Julien et al., 2019) should be replaced by (Moukthar et al., 2019). I remind you that in this publication, the modeling of cell division is based on the "shortest path rule" conditioned by the passage to the nucleus. The two publications should therefore be weighed in the same paragraph!!!

---

## [Reviewer Report]

*Comments to Author*: Dear Dr Weijers and colleagues, 

We have now received the comments from two expert reviewers on your manuscript. Please find their detailed comments attached. 

Both reviewers agreed on the interest and quality of your review article; appreciated as well structured and written. 

They ask however for minor modifications prior acceptance to publication. 

In addition to their comments, I would like to thank you for this useful synthesis, and I may suggest to cite, in the section of the manuscript addressing live imaging studies in the embryo, a new work, published end of November in QPB, by Dr Ueda: Kimata et al., 2020 "Mitochondrial dynamics and segregation during the asymmetric division of Arabidopsis zygotes" Quantitative Plant Biology , Volume 1 , 2020 , e3. DOI: https://doi.org/10.1017/qpb.2020.4

Furthermore, in the section on gene networks or in the conclusion; you might think relevant to cite the recent preprint by Jonsson and Traas' groups "A multi-scale analysis of early flower development in Arabidopsis provides an integrated view of molecular regulation and growth control" DOI: https://doi.org/10.1101/2020.09.25.313312. This would show an example of integration of morphology, gene networks, modelling, at cellular scale and in 4D, which could be also envisioned in the embryo? 

We would be happy to receive a corrected version of your manuscript when it is ready. 

I thank you again for having submitted your manuscript to Quantitative Plant Biology.

Sincerely

Daphné